# Micrometer-Scale Membrane Transition of Supported Lipid Bilayer Membrane Reconstituted with Cytosol of *Dictyostelium discoideum*

**DOI:** 10.3390/life7010011

**Published:** 2017-03-07

**Authors:** Kei Takahashi, Taro Toyota

**Affiliations:** Department of Basic Science, Graduate School of Arts and Sciences, The University of Tokyo, Tokyo 153-8902, Japan; takahashikei0309@gmail.com

**Keywords:** supported lipid bilayer membrane, phosphatidylinositides, cytosol, *Dictyostelium discoideum*

## Abstract

Background: The transformation of the supported lipid bilayer (SLB) membrane by extracted cytosol from living resources, has recently drawn much attention. It enables us to address the question of whether the purified phospholipid SLB membrane, including lipids related to amoeba locomotion, which was discussed in many previous studies, exhibits membrane deformation in the presence of cytosol extracted from amoeba; Methods: In this report, a method for reconstituting a supported lipid bilayer (SLB) membrane, composed of purified phospholipids and cytosol extracted from *Dictyostelium discoideum*, is described. This technique is a new reconstitution method combining the artificial constitution of membranes with the reconstitution using animate cytosol (without precise purification at a molecular level), contributing to membrane deformation analysis; Results: The morphology transition of a SLB membrane composed of phosphatidylcholines, after the addition of cytosolic extract, was traced using a confocal laser scanning fluorescence microscope. As a result, pore formation in the SLB membrane was observed and phosphatidylinositides incorporated into the SLB membrane tended to suppress pore formation and expansion; Conclusions: The current findings imply that phosphatidylinositides have the potential to control cytoplasm activity and bind to a phosphoinositide-containing SLB membrane.

## 1. Introduction

Reconstitution approaches for cell motility have drawn much attention over recent decades. Prof. van Oudenaarden’s group reported that lipid vesicles coated with the *Listeria monocytogenes* virulence protein ActA, are propelled by in vitro actin polymerization [1]. Prof. Takeuchi’s group developed self-propelled micrometer-sized polystyrene beads modified with flagella [2], and Prof. Sonobe’s group demonstrated that an actomyosin fraction containing the sol-gel conversion of a cytoplasmic extract moves like an amoeba [3]. These approaches have been established by using only a fraction of the chemical reaction networks related to the cell machinery. Recently, our group established the membrane-assisted biochemical machinery for amoeba locomotion, using a combination of cytosol extracted from *Dictyostelium discoideum* and lipid film patches [4]. However, these are crude conditions, as the amoeba cell-free extract is simply added to multilamellar lipid patches, whereas real amoeba cells have a cellular membrane with a single lipid bilayer. Therefore, as a next step, the reconstitution of amoeba cells using a single lipid bilayer is required.

A so-called “vesicle rupture method” using small vesicles makes it possible to constitute the unilamellar lipid membrane on a solid surface [5,6,7,8,9,10,11]. This technique is based on the electrical attractive force between the solid glass surface and the lipid, generating a supported lipid bilayer (SLB) [12]. The membrane transformation of the SLB membrane by extracted cytosol from living resources, has recently drawn much attention. For example, there are reports on the membrane transition of purified phospholipid SLB membranes, induced by an injection of cytosol from Escherichia coli. Loose et al. presented observations on the pattern formation and oscillation of the proteins MinD and MinE on an SLB membrane, and produced a reaction-diffusion model of the MinD and MinE dynamics that accounts for this experimental observation [13].

In another report, atomic force microscopy demonstrated that nanometer-sized pores were formed in the pure phospholipid SLB membrane, in the presence of the *Escherichia coli* cytosol [14]. Furthermore, an SLB membrane containing phosphatidylinositol-4,5-bisphosphate can lead to the self-assembly of filopodia-like structures, containing actin bundles with Cdc42, N-WASP-WIP, Toca-1, and the Arp2/3 complex, as well as uniform and short polymerized actin [15]. These reports show that this technique, which we call the novel reconstitution method, using a purified lipid SLB membrane and cytosol from prokaryotic cells or purified proteins related to cell motion or dynamics, is a powerful tool for measuring the energy and kinetics of a membrane transition. Given that the purified lipid SLB membrane is artificially constituted and the extracted cytosol is reconstituted, this current technique is a novel reconstitution method. Here, based on the Noireaux group’s work [14], we develop a new reconstitution method using cytosol extracted from the eukaryotic amoeba *Dictyostelium discoideum*, and by generating a purified phospholipid SLB membrane on a glass substrate using the vesicle rupture method (Figure 1). It enables us to address the question of whether the purified phospholipid SLB membrane. including lipids related to amoeba locomotion, which was discussed in many previous studies, exhibits membrane deformation in the presence of cytosol extracted from amoeba. In this study, the SLB membrane is comprised of purified phosphatidylcholines and phosphatidylinositol phosphates (Scheme 1), and is stained by a fluorescence probe in order to visualize the micrometer-scale membrane deformation in the presence of *Dictyostelium discoideum* cytosol, using a confocal laser scanning fluorescence microscope.

## 2. Materials and Methods

### 2.1. Reagents

1-Palmitoyl-2-oleoyl-3-*sn*-glycero-3-phosphocholine (POPC) was purchased from Wako (Tokyo, Japan). 1-Stearoyl-2-arachidonoyl-*sn*-glycero-3-phospho-(1′-myo-inositol-4′,5′-bisphosphate) (18:0–20:4 PI(4,5)P2), 1,2-hexadecanoyl-3-*sn*-glycerophosphatidylinositol-(3,4,5)-trisphosphate (18:0–20:4 PI(3,4,5)P3), 1,2-dihexanoyl-*sn*-glycero-3-phospho-(1′-myo-inositol-4′,5′-bisphosphate) (08:0 PI(4,5)P2), and 1,2-dihexanoyl-*sn*-glycero-3-phospho-(1′-myo-inositol-3′,4′,5′-trisphosphate) (08:0 PI(3,4,5)P3) were purchased from Avanti Polar Lipids (Alabaster, AL, USA). TexasRed^®^-1,2-dihexadecanoyl-*sn*-glycero-3-phosphoethanolamine triethylammonium salt (TexasRed-DHPE) was provided by Invitrogen (Grand Island, NY, USA). Organic solvents and inorganic salts were provided by WAKO (Tokyo, Japan). These reagents were used as received.

### 2.2. Vesicle Preparation

The phospholipids were dissolved at 0.67 mM with 0.04 mol % TexasRed-DHPE, in 300 μL of a mixed organic solvent (CHCl_3_–MeOH, 2:1, *v*/*v*). The solution was poured into a 10 mL round-bottom flask. The organic solvent was removed for 5 min by a rotary evaporator (N-1110, EYELA, Tokyo, Japan), equipped with a vacuum pump. The speed of the rotation of the evaporator was 180 rpm, the exhaust rate was 1.2 L/min, and the temperature was set to 40 °C by a water bath. After a lipid film was formed at the bottom of the flask, the flask was placed in a desiccator to remove any residual solvent from the lipid film under a reduced pressure at room temperature, for 17 h.

Next, 2 mL of phosphate-buffered saline (PBS) was heated to 37 °C and was gently poured into the round-bottom flask containing the lipid film. The flask was then sealed and incubated at 37 °C for 2 h. The flask was shaken by a vortex mixer, hourly, and for further dispersing, the sample in the flask was ultrasonicated for 1 h. The room temperature was kept at 22 °C. In order to remove undesired tubular vesicles and giant or large vesicles, which are occasionally formed during lipid film hydration, 2 mL of the sample was taken from the vesicle dispersion and passed through a 0.1-μm IsoporeTM Membrane Filter (Merck Millipore, Darmstadt, Germany), twice. This filtering treatment led to small vesicles of phospholipids in PBS.

### 2.3. Cover Glass Washing

First, a cover glass (24 mm × 60 mm, No.1 (thickness = 0.12–0.17 mm), MATSUNAMI, Tokyo, Japan,) was set in a rack (SANSYO, Tokyo, Japan), which was placed in a glass vessel (SANSYO, Tokyo, Japan). A washing solution (Contaminon^®^LS-II (Wako), diluted eight times with MilliQ water (Millipore, Darmstadt, Germany), was poured into the glass vessel and the cover glass was ultrasonicated in the washing solution for 1 h. After ultrasonication, the cover glass was rinsed with distilled water (15 times) and MilliQ water (three times). Further rinsing with MilliQ water (six cycles of ultrasonication for 2 min and further rinsing) was undertaken. Second, ethanol was poured into the vessel and the cover glass was ultrasonicated for 30 min. After the ultrasonication, the cover glass was rinsed with MilliQ water three times. Further rinsing with MilliQ water (six cycles of ultrasonication for 2 min and rinsing ×3) was undertaken again. Third, 0.1 M NaOH(aq) (prepared by MilliQ water) was used as a cleaning solution and the cover glass in the vessel filled with 0.1 M NaOH(aq) was ultrasonicated for 20 min. After ultrasonication, the cover glass was rinsed three times with MilliQ water. Further rinsing with MilliQ water (12 cycles of ultrasonication for 2 min and further rinsing) was undertaken. Finally, the vessel containing the cover glass was heated at 140 °C for 40 min, in a dry atmosphere.

### 2.4. Supported Lipid Bilayer Membrane Preparation

First, the cleaned cover glass was cut to fit a bottom-open dish. After setting up a handmade chamber by attaching the cleaned cover glass to the bottom-open dish with double-sided tape (Frame Seal, thickness; 280 μm, BIO-RAD, Hercules, CA, USA) (Figure 2A), 50 μL of the vesicle dispersion was introduced onto the surface of the cover glass and incubated for 30–60 min at more than 24 °C in an open chamber. Second, the chamber was washed six times by the addition of PBS (200 μL) to the rims of the cover glass, which gently soaked it up, resulting in the removal of surplus vesicles (Figure 2B).

### 2.5. Fluorescence Microscopy Observation

The SLB membrane was observed using a confocal laser scanning fluorescence microscope (CSU-X1, IX81, Olympus, Tokyo, Japan), equipped with a 100× objective lens (N.A. = 1.30, working distance = 0.2 mm). Red and green fluorescence images were obtained by using the corresponding band-pass filter and dichroic mirror units (excitation 520–550 nm, emission >580 nm and excitation 470–490 nm, emission 510–550 nm). For image analysis of the areas with pores in the SLB membrane, binarization processing of the brightness was performed on the microscopy images, to clarify the information on the edges and the coordinates of the center of mass using the image analysis software ImageJ (NIH). We calculated the area of the micrometer-sized structures in pixel × pixel unit. The area unit was converted from pixel × pixel to μm^2^, using a graduated glass-scale plate.

### 2.6. Cell Culture

AX4 cells co-expressing the Pleckstrin homology domain of cytosolic regulator of adenylyl cyclase (PH-crac) tagged with RFP (PH-crac-RFP), and phosphatase and tensin homolog deleted from chromosome 10 (PTEN) and tagged with GFP (PTEN-GFP), were cultured in modified HL5 medium containing G418 (30 μg/mL, Wako) and Hygromycin B (60 μg/mL, Calbiochem) [16], under shaking at 155 rpm at 22 °C [17].

### 2.7. Cytosol Extraction

Cytosolic extracts were prepared from AX4 cells, according to the protocol described elsewhere [18,19]. The cells were harvested at a cell density of 7 × 10^6^ cells mL^−1^, washed two times with 80 mL of phosphate buffer (PB), and re-suspended in 500 μL of PB. Cells were disrupted by nitrogen decompression using a cell disruption vessel (model 4639, Parr Instrument Co., Moline, IL, USA) on ice. The cell lysate was centrifuged at 16,000× *g* for 30 min at 4 °C. The supernatant was isolated and centrifuged again. The resulting supernatant was transferred to a microtube and kept on ice until just before use.

## 3. Results and Discussion

### 3.1. Supported Lipid Bilayer Membrane Formation by Rupturing Vesicles on Glass

In order to trace the formation of an SLB membrane from vesicles stained with 0.04 mol % TexasRed-DHPE, timelapse image capture using a confocal laser scanning fluorescence microscope was performed. The confocal system was focused on the surface of the cover glass of the chamber after 50 μL of 0.1 mM POPC vesicle dispersion was gently injected onto the glass surface. As shown in Figure 3A, the glass surface was covered by the fluorescent membrane for the initial 600–1020 s after the vesicle dispersion injection. The vesicle rupture method for the formation of an SLB membrane enables us to generate heterogeneous films composed of a lipid mixture [5,6,7,8,13,14]. When the vesicle dispersion of POPC and 2 mol % phosphatidylinositide was dropped on the glass surface, the fluorescent membrane was formed in the same manner as POPC vesicles in the initial 1020–1500 s after the vesicle dispersion injection. In order to remove the surplus vesicles, the chamber was washed more than 12 times with PBS (Figure 2B).

After washing the chamber, which was incubated for 30 min after the vesicle dispersion injection, the fluorescence intensity profile (the mean fluorescence intensity of an 81 × 81 μm^2^ area) along the z-position in red modes (excitation 520–550 nm, emission > 580 nm) revealed that a single peak with a full width at half maximum, of less than 1 μm, was located at the z position of the surface of the cover glass (Figure 3B,C). Thus, the vesicles in the dispersion were transformed to the SLB membrane on the glass substrate, via rupturing in the PBS [20,21]. We also observed the POPC-based SLB membrane, including the phosphatidylinositol phosphates, which revealed a single peak with a full width at half maximum, of less than 1 μm, similar to the SLB membrane for only POPC.

### 3.2. Pore Formation on the SLB Membrane after Injection of Cytosol Extract

Cytosol extracted from *Dictyostelium discoideum* (the PH-crac-RFP/PTEN-GFP co-expressing strain) was injected onto the SLB membrane formed in the chamber. As shown in Figure 4A and Video S1, pore formation in the POPC SLB membrane was observed in the initial 444–552 s after the injection of cytosol. Then, every pore swelled in the x-y plane, almost at the same time. As a negative control reference in terms of the ionic strength change of the buffer and protein degeneration, pore formation did not occur in the POPC SLB membrane after the injection of previously boiled cytosol (110 °C, over 1 h). Figure 4B shows the time-course change of the standard deviation of red fluorescence intensity (an area of 81 × 81 μm^2^) of the POPC SLB membrane. The effect of the injection of cytosol, i.e., the increase of the standard deviation, was observed in the initial 60 s, possibly because the fluorescence intensity increased by adsorption of fluorescent proteins. At around 240 s, brilliantly fluorescent spots disappeared, affording the transient change of S.D. They plausibly correspond to aggregate residues of vesicles or localized multiple layers of membranes attached to the SLB membrane and detached to the cytosol. The arrow depicted in Figure 4B shows the transition of the SLB membrane. From 400 s to 600 s, the standard deviation tended to increase, whereas the mean of the fluorescence intensity gradually decreased, due to bleaching. This suggests an increase in the variation of the fluorescence intensity of the SLB membrane due to pore formation (dark spots on the fluorescent image). Figure 4C shows the profile of the mean fluorescence intensity of the green mode along the z position. Before the injection of cytosol, the fluorescence intensity of the SLB membrane was negligible. The peak was found at the glass surface, whereas a certain value of the fluorescence intensity was detected over the glass surface. This result indicates that the green fluorescent molecule, PTEN-GFP, was localized to the POPC SLB membrane after the injection of cytosol. We examined the cytosol of the wild AX4 cells for injection and observed the pore formation of the POPC SLB membrane on a micrometer scale, in the presence of cytosol. To examine the temperature effect, this experiment was performed at low room temperature (21 °C) (note that all other experiments were carried out at 24–26 °C). The pore formation in the POPC SLB membrane was observed in the initial 1632–3000 s after the injection of cytosol (Appendix A). The decrease in the temperature caused a delay in pore formation in the POPC SLB membrane below the cytosol.

The results of pore formation indicated that the cytosol extracted from *Dictyostelium discoideum* has the potential to induce membrane transitions of the SLB membrane composed of only POPC. In fact, the SLB membrane deformation on a nanometer scale has been reported elsewhere. For instance, the atomic force microscopy observation revealed that the formation of nanometer-sized pores in the 1,2-dimyristoyl-*sn*-glycero-3-phosphocholine SLB membrane was caused by a poly-l-lysine injection [22]. *Escherichia coli* cytosol expressing the pore-forming membrane protein α-hemolysin, induced pore formation in POPC SLB membranes on a nanometer scale, observed by AFM [14]. These studies imply that proteins or cytosol cause a SLB membrane of phospholipid to form pores. Figure 4A,B shows that cytosol extracted from *Dictyostelium discoideum* also has the potential to deform membranes. We established a micrometer-sized experimental model that demonstrated membrane deformation using amoeba cytosol.

### 3.3. Disturbance of Pore Formation in the SLB Membrane by Phosphatidylinositides

Figure 5A shows the time-course change of fluorescence microscopy images of an SLB membrane containing POPC and 2 mol % 18:0–20:4 PI(4,5)P2, after the injection of cytosol. The SLB membrane exhibited pore formation in the initial 720–900 s after the injection of cytosol and the pores expanded in the x-y plane. In contrast, the SLB membrane of POPC and 2 mol % 18:0–20:4 PI(3,4,5)P3 did not change after the injection of cytosol at 24–30 s (Appendix A). No pores developed in the SLB membrane, but instead, a slight lipid aggregation was observed. The PTEN-GFP localization in both SLB membranes containing phosphatidylinositides was confirmed by the green mode profile of the fluorescence intensity along the z-position after the injection of cytosol (Appendix A). Figure 5B shows the time-course change of the standard deviation of red fluorescence intensity for the SLB membrane composed of POPC/18:0–20:4 PI(4,5)P2. The arrow points to the pore formation in the POPC/18:0–20:4 PI(4,5)P2 SLB membrane, occurring from 720 s to 900 s. The standard deviation tended to increase in a similar manner to that of the POPC SLB membrane. Moreover, the distribution of the minimum value of distance (the center of mass) between the centroids of pores in the captured images of the POPC SLB membrane and the SLB membrane of POPC/18:0–20:4 PI(4,5)P2, was evaluated by image analysis. The distribution in both cases was in the range of 3.3–7.3 μm. The nearest neighbor distance between the centroids of the pores did not change when the SLB membrane contained 18:0–20:4 PI(4,5)P2.

The length of the acyl chain of phospholipids correlates with the fluidity of the bilayer membrane. Hence, the 08:0 PI(4,5)P2 and 08:0 PI(3,4,5)P3, which have shorter acyl chains than 18:0–20:4 PI(4,5)P2 and 18:0–20:4 PI(3,4,5)P3, were also examined when mixed into the POPC SLB membrane. Appendix AA shows the fluorescence microscopy images of the SLB membrane containing POPC and 2 mol % 08:0 PI(4,5)P2 before, and 30 min after, the injection of cytosol. The pores were formed in the SLB membrane and expanded in the x-y plane. When using 2 mol % 08:0 PI(3,4,5)P3, pore formation was also observed after the injection of cytosol (Appendix AB). The areas of pores were measured using the threshold for adjusting the binarization to the fluorescence microscopy image.

Appendix AC shows the binarization of the POPC SLB membrane with pores 20 min after the injection, whereas Figure 5C depicts the diagram of the areas of the pores in each SLB membrane, observed 25 min after the cytosol injection. The pore size in the POPC SLB membrane was larger than that of any other SLB membrane. This analysis implies that the SLB membranes based on the POPC including phosphatidylinositides, tended to disturb pore formation or expansion. The POPC/18:0–20:4 PI(3,4,5)P3 SLB membranes, especially, exhibited no pore formation after the cytosol injection. Therefore, the potential of cytosol to deform membranes in a reconstitution system is controlled by phosphoinositides.

Note that the pores formed in the POPC SLB membrane expanded to a micrometer size. Because PTEN is a well-known membrane-binding protein, specifically localizing to 18:0–20:4 PI(4,5)P2 [23], it is important to examine the cytosol injection to the SLB membrane including phosphoinositides, to approach the mechanism of pore formation. Moreover, pore formation was sensitive to temperature (Appendix A). Two reasons why the temperature affected the initial timing of the pore formation can be envisioned: (i) the membrane fluidity and (ii) the activity of membrane-binding proteins. The membrane fluidity becomes low as the temperature decreases, especially when the membrane transition from a liquid crystal phase (fluid) to a gel phase (solid-like) occurs [24]. However, POPC has a phase transition temperature of about 0 °C; whereas, the temperature of the experimental conditions mentioned above was much higher than 0 °C. Hence, the activity of membrane-binding proteins is possibly a determinant of the pore formation in the SLB membrane observed here.

In the presence of cytosol, the SLB membranes also contained tubular giant vesicles (tGV). In spite of the same chemical conditions as shown in Figure 4, tGVs were formed in the POPC SLB membrane on the surface of the cover glass, without the bottom-open dish (Appendix A, Video S2). In this condition, the pores which formed in the SLB membrane were smaller than those observed in Figure 4A, at 30 min after the injection of cytosol. The bottom-open dish used for the chamber appears to resist the slight distorting force of the cover glass surface. Thus, the distortion-free surface may only allow pore formation [25]. The formation time of the pores and tGVs, as well as their sizes, are similar, not to phagocytosis, because phagocytosis-inducing foreign particles or cells are absent in the current experiment, but to macropinocytosis of amoeba. The model describing the regulation of macropinocytosis in *Dictyostelium discoideum* depicts the requirement of the activity of RasS, PIK1, PIK2, and PKB, to complete the formation of a macropinosome on a micrometer scale, for several tens of seconds [26,27,28]. Recently, some reports using *Dictyostelium discoideum* demonstrated that macropinocytosis is regulated by a network of Ras family members [29] or the level of phosphatidylinositol 3,4,5-trisphosphate [30,31] in vivo. This model implies that cytosol extracted from *Dictyostelium discoideum* affects the pure lipid SLB membrane, resulting in pore formation, which is inhibited by phosphatidylinositides, and the formation of tGVs.

## 4. Conclusions

In this report, SLB membranes composed of POPC and phosphatidylinositides were constructed through the vesicle rupture method. The injection of cytosol extracted from *Dictyostelium discoideum* onto the SLB membrane induced pore formation in the SLB membrane. Pore formation was inhibited by phosphatidylinositide. These results imply that cytosol has the potential to induce membrane transitions on a micrometer scale and that phosphatidylinositide tunes membrane deformation. We established a reconstitution method using amoeba cytosol for revealing micrometer-sized membrane deformations. This technique is a powerful in vitro tool for the investigation of membrane deformation in terms of cell motion and intercellular motility.

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
