# Peer review of "Micrometer-Scale Membrane Transition of Supported Lipid Bilayer Membrane Reconstituted with Cytosol of Dictyostelium discoideum"

_life, 2017, doi:10.3390/life7010011_

Reviewer 1 Report

Takahashi and Toyota report morphological transitions of supported lipid bilayer (SLB) membranes composed of purified phosphatidylcholine lipids upon the introduction of extracted cytosol components. It was discovered that the cytosol components induce pore formation in the SLB membrane, whereas the presence of phosphatidylinositides in the SLB membrane hindered pore formation and expansion. Overall, this is a very nice study that extends the capabilities of supported lipid bilayers to reconstitute amoeba cells. As the authors appear to have deep expertise in studying amoeba biology, I would like to offer a few suggestions below about giving the manuscript proper context to the supported lipid bilayer field as well because the study also advances other recent work involving the interaction of purified membrane-active compounds with supported lipid membranes and resulting morphological changes in the membrane structure, including tubule and budding. These minor revisions will greatly improve the manuscript and I recommend that the manuscript be published in Life after these revisions are made:

1. In Paragraph 2 of the Introduction, it would be good to extend the motivation to add more references that discuss how supported lipid bilayers are utilized to study morphological transitions with compound addition. I recommend that his information should be placed between the introduction of SLB and mention of recent studies on cytosol addition to SLBs. Right now, the paragraph switches from SLBs in general to the cytosol subject and there should be more discussion in between, especially the following relevant references for SLB morphological transitions should be added: DOI: 10.1021/la700523x and DOI: 10.1021/acs.langmuir.5b02088 and DOI: 10.1103/PhysRevLett.110.028101 

2. In Figure 3, why are the fluorescence intensity profiles of the fabricated SLBs "spotty". Typically, good-quality SLBs have a smooth fluorescence intensity profile. Please clarify and address how the thresholding was performed and if you have any additional verification for complete SLB formation. Of note, the SLB quality looked better in Fig. 4A. 

3. It is mentioned that "As a negative control reference, pore formation did not occur in the POPC SLB membrane after injection of previously boiled cytosol." Why does this happen? Does this mean pore formation is mediated by a cytosolic component that undergoes irreversible temperature denaturation (e.g., protein) or does the boiling cause a simple aggregation of cytosolic components rendering them inoperable? Please clarify and discuss the possibilities in the manuscript.

Author Response

We greatly appreciate the insightful comments from the  reviewers on our manuscript. We have addressed these comments one-by-one and believe that the revised version becomes suitable for publication. Our detailed responses to the comments are outlined below. We attached the revised manuscript as the word file which the revised sites are highlighted in red color.

1.    In Paragraph 2 of the Introduction, it would be good to extend the motivation to add more references that discuss how supported lipid bilayers are utilized to study morphological transitions with compound addition. I recommend that his information should be placed between the introduction of SLB and mention of recent studies on cytosol addition to SLBs. Right now, the paragraph switches from SLBs in general to the cytosol subject and there should be more discussion in between, especially the following relevant references for SLB morphological transitions should be added: DOI: 10.1021/la700523x and DOI: 10.1021/acs. langmuir.5b02088 and DOI: 10.1103/PhysRevLett.110.028101

----We appreciate reviewer 2 for pointing the important papers. We have added them to the reference as [9]-[11] in the revised version

2.    In Figure 3, why are the fluorescence intensity profiles of the fabricated SLBs "spotty". Typically, good-quality SLBs have a smooth fluorescence intensity profile. Please clarify and address how the thresholding was performed and if you have any additional verification for complete SLB formation. Of note, the SLB quality looked better in Fig. 4A.

----Figure 3 shows the SLB membrane formed by vesicle addition before rinse. As a result, the brilliantly fluorescent spots appeared and they plausibly correspond to aggregate residues of vesicles or localized multiple layers of membranes associated with the SLB membrane formation. Most of them were rinsed in the last step of the preparation of SLB membrane as the reviewer has pointed out in Fig. 4A. Further verification on SLB membrane was difficult because our current observation setup focused on observing the sample of the micrometer-sized pores formed randomly and simultaneously with the large field of view. Practically speaking, the disappearance of brilliantly fluorescent spots was observed and the gradual pore formation started around 3 minutes after the disappearance. This induction time implies that the start of pore formation and the disappearance of the brilliantly fluorescent spots have no direct relationship with each other.

3.    It is mentioned that "As a negative control reference, pore formation did not occur in the POPC SLB membrane after injection of previously boiled cytosol." Why does this happen? Does this mean pore formation is mediated by a cytosolic component that undergoes irreversible temperature denaturation (e.g., protein) or does the boiling cause a simple aggregation of cytosolic components rendering them inoperable? Please clarify and discuss the possibilities in the manuscript.

----The boiled cytosol was prepared by the incubation at 110 oC over 1 hour at atmospheric pressure and the negative control reference was assigned to the ionic strength change of the buffer and it includes the protein unfolding or aggregation. We have explained it in the revised version.

Reviewer 2 Report

The authors present a method for studying the effect of Dictyostelium discoideum cytosol on pre-reconstituted supported lipid bilayer of several phospholipid compositions. The lipid formulations analyzed are biologically relevant as they contain, in addition to phosphatidylcholine, also mixtures made with phosphatidylinositide molecules. Thus, mimicking the composition of real amoeba single lipid bilayer. The experiments are well executed coupling a novel method for making supported lipid bilayer (SLB) to confocal microscopy observations and analysis with cellular extracts. The goal of the authors was to report a quite novel experimental method, and assess the lipid composition responsible of single bilayer deformation upon injection of Dictyostelium discoideum cytosol. The results are clear and of potential interest to scientists of different fields from biophysics, synthetic biology, origin and artificial life. I support publication of the work after minor revision.     

The comments below may help to improve the quality of the work, and facilitate understanding of a broader scientific audience.

Line 32: Please state why reconstitution approaches have drawn more attention recently.

Line 48: Why is interesting to study such phenomena?

Line 57 to 62: Please try to write a more sharp definition of why you called this method “pre-reconstitution” method. In my opinion your definition is not clear and convincing, i.e. I don’t see the connection between the description given and the term chosen to define it. Make it clearer. Also in the abstract why you describe the cytosol as “animate”? Please, explain it in the test.

Line 105: Please state why the cover glass is an important step in the preparation method.

Line 129: Please report the numerical aperture and working distance of the objective used in this study with the bottom-open dish.

Line 143: Please state why this specific cell line was chosen for your experiments. If you prefer you may add this information in the introductory part of your manuscript. In particular, explain why PH-crac and PTEN were co-expressed.

Line 146: Check the concentration of Hygromycin B.

Line 149: Please report the total protein concentration in mg/mL of the AX4 cytosolic extract. If needed perform a Bradford assay to determine the concentration. This parameter could be important for others to reproduce the results reported.

Line 162: Correct 1200 with 1020 seconds.

Line 176: Please show (supporting material) micrographs and single pick image analysis result of POPC based SLB.

Line 192: State what you expect when the cytosol is boiled.

Line 238: Please show the green profile along the z-position after injection of the cytosolic extract (Supporting material).

General remarks:

Please add a table with the chemical formula of the lipids used in this studies and report in it the experimental observation related to the lipid molecule used. In the conclusion give a plausible explanation of why phosphoinositides tend to prevent pore formation, and which could be the factor/s (e.g. proteins) in the cell cytoplasm inducing the pore formation. Also interesting the observation of tGVs associated to the macropinocytosis phenomena – maybe emphasize it in the conclusions.   

Author Response

We greatly appreciate the insightful comments from the  reviewers on our manuscript. We have addressed these comments one-by-one and believe that the revised version becomes suitable for publication. Our detailed responses to the comments are outlined below. We attached the revised manuscript and supplementary materials as the word file which the revised sites are highlighted in red color.

Line 32: Please state why reconstitution approaches have drawn more attention recently.

----The reason is the development of synthesis and handling of biological molecules and nanometer-sized assembly technique of such molecules in the laboratories.

Line 48: Why is interesting to study such phenomena?

----Because we can investigate the membrane deformation and trace them in real-time speed under stimuli including cytosol. As pointed out by the reviewer 2 and 3, we added references [9]-[11] in the revised version to emphasize this viewpoint.

Line 57 to 62: Please try to write a more sharp definition of why you called this method “pre-reconstitution” method. In my opinion your definition is not clear and convincing, i.e. I don’t see the connection between the description given and the term chosen to define it. Make it clearer. Also in the abstract why you describe the cytosol as “animate”? Please, explain it in the text.

----According to reviewer 3’s comment, we revised “pre-reconstituition” to “new” reconstituition in main text including the title. “Animate cytosol” means the cytosol without precise purification in molecular level. We have added this explanation to the abstract.

Line 105: Please state why the cover glass is an important step in the preparation method.

Line 129: Please report the numerical aperture and working distance of the objective used in this study with the bottom-open dish.

----In our current setup, we used the maximum NA (1.30) and the highest-magnification lens (100x). The working distance is estimated to be around 0.2 mm. Thus we need to use the cover glass the thickness of which was 0.12-0.17 mm. We have reported these information in the revised version.

Line 143: Please state why this specific cell line was chosen for your experiments. If you prefer you may add this information in the introductory part of your manuscript. In particular, explain why PH-crac and PTEN were co-expressed.

----We examined the wild AX4 cells and obtained the SLB membrane with pores under the cytosol injection. In the current study, we aimed to visualize the localization of PTEN to the SLB membrane and used AX4 cells co-expressing PH-crac-RFP and PTEN-GFP which was kindly gifted from Prof. Sawai (The University of Tokyo). We have revised the text.

Line 146: Check the concentration of Hygromycin B.

Line 162: Correct 1200 with 1020 seconds.

----We thank reviewer 3 for pointing out them. We have revised both.

Line 149: Please report the total protein concentration in mg/mL of the AX4 cytosolic extract. If needed perform a Bradford assay to determine the concentration. This parameter could be important for others to reproduce the results reported.

---- We did not examine total protein evaluation but controlled the cell number (0.7x108 cells in 500 mL PB buffer) to prepare the cytosol for each injection and obtained reproducible results showing pore formation of the SLB membrane.

Line 176: Please show (supporting material) micrographs and single pick image analysis result of POPC based SLB.

---- We raised Figure S4C for the reviewer’s request.

Line 192: State what you expect when the cytosol is boiled.

----The boiled cytosol was prepared by the incubation at 110 oC over 1 hour at atmospheric pressure and the negative control reference was assigned to the ionic strength change of the buffer and it includes the protein unfolding or aggregation. This statement has been added in the revised version.

Line 238: Please show the green profile along the z-position after injection of the cytosolic extract (Supporting material).

----According to reviewer 3’s comment, we added the green fluorescence intensity profile as Figure S3.

General remarks:

Please add a table with the chemical formula of the lipids used in this studies and report in it the experimental observation related to the lipid molecule used. In the conclusion give a plausible explanation of why phosphoinositides tend to prevent pore formation, and which could be the factor/s (e.g. proteins) in the cell cytoplasm inducing the pore formation. Also interesting the observation of tGVs associated to the macropinocytosis phenomena – maybe emphasize it in the conclusions.

----We appreciate reviewer 3 for advising us for reinforcing the manuscript. Chemical formula were added as Scheme 1 in the text. It is certain that the observation of tGVs associated to the macropinocytosis phenomena is interesting, but we focused on the pore formation of SLB membrane under cytosol, thus we simply remarked the role of phosphatidylinositides in the conclusion.

Reviewer 3 Report

The article by K. Takahashi and T. Toyota represents a sound study on a remarkable phenomenon: a structural transformation of a lipid bilayer connected to the formation of pores induced by extracted cytosol. All procedures are well described, the protocol of the observations seem sound and convincing.

 However, there are a number of issues of some importance which should be clarified or corrected by the authors:

1) Chapter 2.2 “Vesicle preparation”: it would be nice to know something about the size of the vesicles: has it been determined? Are the vesicles visible on the scale of an optical microscope? This information would also help to interpret the initial (0 sec) features shown in Figs. 3-5.

 2) In Fig. 3 and the relating text, the authors put emphasis on the fluorescence intensity profile (Fig. 3 B) and the fact that its width is below one micrometer. However, given that the cross section of a bilayer membrane is only something like 5 nm, this would be a very large value. Is this approach capable of telling a single bilayer from a large stack of bilayers (multilayer)? This question should be discussed.

 3) In Fig. 4 B, a peak of the standard deviation around 240 sec is observed. No corresponding structures are shown in the micrographs in Fig. 4 A, the critical time window is missing. It would be interesting to take a closer look at a micrograph at 240 sec to identify the source of the temporarily larger SD.

 4) In all micrographs, the pores exhibit a distinctly irregular shape, none of them appears to be circular or elliptical. Would it be possible to take a closer look at these pores with a higher resolution? Given that the displayed square has a side length of 75.5 micrometers, this should be doable. Resulting images could reveal valuable information on the nature of the pores.

Minor issues:

a) Line 35: “polystylene” should probably read “polystyrene”.

b) Line 291: skip one “at”

c) Line 294: “absence” should be replaced by “absent”.

Author Response

We greatly appreciate the insightful comments from the  reviewers on our manuscript. We have addressed these comments one-by-one and believe that the revised version becomes suitable for publication. Our detailed responses to the comments are outlined below. We attached the revised manuscript as the word file which the revised sites are highlighted in red color.

1)    Chapter 2.2 “Vesicle preparation”: it would be nice to know something about the size of the vesicles: has it been determined? Are the vesicles visible on the scale of an optical microscope? This information would also help to interpret the initial (0 sec) features shown in Figs. 3-5.

---- We used 0.1-mm carbonate filters to reduce the diameters of vesicles and observed the membrane with the large field of view for observation of micrometer-sized pores formed simultaneously after cytosol injection. Thus each vesicles were not observable since their diameters were less than the length of each pixel and diffraction limit. The brilliantly fluorescent spots on the SLB membrane observed at 0 sec in figures are assigned to aggregates of vesicles or localized multiple layers of membranes associated with the SLB membrane formation.

2)    In Fig. 3 and the relating text, the authors put emphasis on the fluorescence intensity profile (Fig. 3 B) and the fact that its width is below one micrometer. However, given that the cross section of a bilayer membrane is only something like 5 nm, this would be a very large value. Is this approach capable of telling a single bilayer from a large stack of bilayers (multilayer)? This question should be discussed.

---- Our current observation setup focused on observing the sample of the micrometer-sized pores formed randomly and simultaneously with the large field of view. Thus, instead of precise information on the thickness of SLB, we raised the experimental fact that the membrane width (thickness) was below one micrometer and the references 21-23 to indicate that the membrane is formed through vesicle ruptures. “Spotty” fluorescence microscopy images indicate that there were aggregate residues of vesicles or localized multiple layers of membranes associated with the SLB membrane formation. Most of them were rinsed in the last step of the preparation of SLB membrane. Therefore we consider that such issue of SLB membrane does not interfere the role of phosphatidylinositols on the pore formation. We have added this interpretation in the revised version.

3)    In Fig. 4 B, a peak of the standard deviation around 240 sec is observed. No corresponding structures are shown in the micrographs in Fig. 4 A, the critical time window is missing. It would be interesting to take a closer look at a micrograph at 240 sec to identify the source of the temporarily larger SD.

---- It is true that the brilliantly fluorescent spots disappeared at 240 sec, affording the transient change of SD. They plausibly correspond to aggregate residues of vesicles or localized multiple layers of membranes associated with the SLB membrane formation. Their disappearance was observed and the gradual pore formation started around 3 minutes after the disappearance. This induction time implies that the start of pore formation and the disappearance of the brilliantly fluorescent spots have no direct relationship with each other.

4)    In all micrographs, the pores exhibit a distinctly irregular shape, none of them appears to be circular or elliptical. Would it be possible to take a closer look at these pores with a higher resolution? Given that the displayed square has a side length of 75.5 micrometers, this should be doable. Resulting images could reveal valuable information on the nature of the pores.

---- We appreciate the reviewer for pointing out another discussion viewpoint about the pore formation. However we cannot approach the precise high-resolution microscopy images with the current setup because we used the maximum NA (1.30) and the highest-magnification lens (100x) in them.

Minor issues:

a) Line 35: “polystylene” should probably read “polystyrene”.

b) Line 291: skip one “at”

c) Line 294: “absence” should be replaced by “absent”.

----We thank reviewer 1 for pointing the minor issues. We have revised all of them.

Round  2

Reviewer 1 Report

The authors have satisfactorily replied to the comments made by myself and the other Reviewer. I recommend that the manuscript is now acceptable for publication. 

Reviewer 2 Report

The authors took care of answering my comments and suggestions. The revised version of the manuscript is improved. 

Reviewer 3 Report

I feel that the authors have sufficiently dealt with the issues I brought up in my first review. To my opinion, the manuscript is now suitable for publication.